# Clinical Outcome after Medial Open-Wedge High Tibial Osteotomy: Comparison of Two Angular Stable Locking Plates—TomoFix™ versus LOQTEQ^®^ HTO Plate

**DOI:** 10.3390/jpm13030472

**Published:** 2023-03-05

**Authors:** Hi-Un Park, Henrik Constantin Bäcker, Martin Häner, Karl F. Braun, Wolf Petersen

**Affiliations:** 1Department of Orthopaedic and Trauma Surgery, Martin-Luther-Hospital, 14193 Berlin, Germany; 2Charité University Hospital Berlin, 10117 Berlin, Germany; 3Department of Orthopaedic Surgery and Traumatology, Auckland City Hospital, Auckland 1023, New Zealand; 4Klinikum Rechts der Isar, Klinik und Poliklinik für Unfallchirurgie, Technische Universität München, 81675 München, Germany

**Keywords:** open-wedge HTO, osteosynthesis, internal plate fixator, angular stability, pseudarthrosis, osteoarthritis, clinical outcome

## Abstract

This study evaluated bony healing and clinical results after medial open-wedge HTO to compare the outcome of the LOQTEQ^®^ HTO plate and the TomoFix™ internal plate fixator. A prospective, non-randomised, comparative study was undertaken. The same surgical technique for the medial open-wedge HTO was used in two treatment groups. In Group 1, the TomoFix™ implant was used for osteosynthesis, and, in Group 2, the LOQTEQ^®^ HTO plate was used. All patients were examined before surgery (T0) and then at 12 months (T1) and at 24 months (T2) postoperatively. The primary outcome measure was the KOOS pain subscore. The secondary outcome criteria were other KOOS subscales, the Tegner score, radiological healing (RUST), and incision length. The KOOS pain subscale and the other KOOS subscores increased significantly in both groups from T0 to T1 and T2 without a significant group difference at each timepoint. The activity measured with the Tegner scale increased significantly from T0 to T2 without a significant group difference. No radiological signs of implant failure were observed in any case at the one-year X-ray, and no patient fulfilled the criteria for non-union. There was no significant difference in the frequency of adverse effects between the two treatment groups. The length of the incision was significantly shorter in the LOQTEQ^®^ HTO group than in the TomoFix™ group. The results of this study show that patient-related outcome scores (KOOS, Tegner) increased after medial open-wedge HTO. There was no difference in clinical outcome or radiological healing between the treatment groups. Both plates are suitable for the osteosynthesis of open-wedge HTO.

## 1. Introduction

High tibial osteotomy (HTO) is a method for the treatment of unicompartmental osteoarthritis in combination with varus deformity [1,2,3,4,5,6]. The most commonly used surgical techniques are open- or closed-wedge osteotomy [1,2,3,4,5,6]. Open-wedge osteotomy is considered the gold standard in lowering the risk of peroneal nerve injuries. The amount of correction can be accurately adjusted intraoperatively [7].

Medial open-wedge HTO is, however, an unstable condition for the proximal tibia, and so a stable fixation device is needed to stabilise the open wedge osteotomy and improve bone union [7].

There are different plate types for the stabilisation of open-wedge HTO: locking vs. non-locking plates and short spacer plates vs. rigid, long plates. Applying short plates is a simple procedure which is commonly used. Some authors have shown that the use of short spacer plates for the stabilisation of open-wedge HTO is associated with a high plate-associated complication rate such as rate of non-union and implant failure [5,8]. High plate-associated complication rates have even been described for implants with locking screws (e.g., Position Plate^®^; Aesqulap, Germany) [9,10,11]. Most authors, therefore, favour long, rigid implants with locking screws for the internal fixation of a medial open-wedge osteotomy [1,2,3,4,5,6,12,13]. The most popular implant is a long spacer plate with multi-directional locking screws (TomoFix™, Synthes, Switzerland). This implant yielded the good results in biomechanical and clinical studies [3,4,14]. There was a low complication rate for open-wedge osteotomy stabilised with the TomoFix™, and this was mostly related to surgical causes [4]. One disadvantage of the TomoFix™ implant is its prominent size. The implant is especially visible in smaller or thin patients and may lead to discomfort during the early rehabilitation phase. Niemeier et al. found that 40% of patients experienced plate-related pain after HTO stabilised with the TomoFix™ implant [4].

This disadvantage led to the development of a new, shorter fixation device (LOQTEQ^®^ HTO plate, aap Implants, Germany). The LOQTEQ^®^ HTO plate is 90 mm long versus the 115 mm length of the TomoFix™ plate. A biomechanical study has shown that, despite its smaller size, the material (Ti_6_Al_4_V alloy) results in high mechanical stability and a trapezoidal plate profile (Figure 1A) [15]. Biomechanical data should be translated into clinical practice with caution, however.

Unfortunately, clinical data for the LOQTEQ^®^ HTO plate are limited and are urgently required in order to assess the safety of the new implant. This study, thus, examined the clinical outcomes of this new osteosynthesis system in comparison with the clinically proven benchmark TomoFix™ (Figure 1B).

The hypothesis of this study was that medial open-wedge HTO with the LOQTEQ^®^ HTO plate provides better pain relief due to the smaller implant size compared to a medial open-wedge HTO with the larger TomoFix™ implant. In the early postoperative period, the pain is mostly related to the osteotomy, whereas this changes in the long term. In the long term, the rationale for this hypothesis was that the smaller implant size contributes to less plate-related irritation and, subsequently, pain. A low rate of osseous healing complications was expected for the LOQTEQ^®^ HTO plate because a previous biomechanical study showed that the primary stability of both implants is comparable [5].

## 2. Materials and Methods

### 2.1. Patients

The patient recruitment and baseline data collection for this prospective, non-randomised but comparative cohort study was undertaken at our hospital between 1 January 2015 and 31 December 2017.

Thirty-two patients who received a TomoFix™ implant between 1 January 2016 and 31 October 2016 were included in the present study. Thirty-five patients who received a LOQTEQ^®^ HTO plate between 1 November 2016 and 31 September 2017 were included. Inclusion and exclusion criteria are shown in Table 1.

The study design was explained to patients who fulfilled all inclusion criteria. If an appropriate patient gave informed consent, then they were provisionally included. Final inclusion was determined intraoperatively when cartilage status was assessed.

The study design was approved by the medical ethics committee of the medical faculty of the Charité-Universitätsmedizin Berlin (no. EA4/055/15).

### 2.2. Surgical Technique

The same surgical technique for the medial open-wedge HTO was used in both groups [11], and the only difference was the implant used for the osteosynthesis (Group 1: TomoFix™; Group 2: LOQTEQ^®^ HTO plate). Preoperative, computer-based planning of the osteotomy was performed using the method described by Miniaci [16] and Medicad^®^ software. All patients were operated on by the same surgeon.

The surgical procedures were performed under general or spinal anaesthesia in both groups. A perioperative antibiotic prophylaxis was performed with cefuroxime (1.5 g). An arthroscopy was performed to evaluate the cartilage status and treat other intraarticular pathological findings. The intraoperative status of the cartilage was documented according to the IKDC documentation form (0: normal; Grade I: nearly normal, superficial lesions; Grade II: abnormal, cartilage defects extending to 50% of cartilage depth; Grade III: severely abnormal, cartilage defects extending to more than 50% of cartilage depth; Grade IV: severely abnormal, cartilage lesions extending to the subchondral bone plate and deeper). Lateral torn menisci were partially removed. The loss of meniscus tissue was documented as a percentage of the non-injured meniscus. A notchplasty was performed in cases involving a narrow notch with osteophytes.

A vertical anteromedial approach was used for the medial open-wedge HTO. After exposing the pes anserinus superficialis, a Hohmann retractor was placed in the infrapatellar bursa. A rasp was used to detach the superficial portion of the medial collateral ligament from the tibial bone, then a second Hohmann retractor was placed behind the posterior cortex of the proximal tibia. The oblique osteotomy was marked with two 2.0 mm K-wires with the help of an image intensifier. The entry point of the K-wires was just above the medial hamstring tendons. The hinge of the osteotomy was in the upper part of the tibiofibular joint. Once the K-wires were in the correct position, the osteotomy was performed with an oscillating saw (Synthes, Raynham, MA, USA, TRS Modular Drive). During the osteotomy, the saw was cooled with fluid (NaCl) to avoid heat damage to the tibial bone. The osteotomy saw stopped 5–10 mm before the lateral cortex to leave the lateral hinge intact, then the osteotomy was spread with three Lambotte chisels and a spreader. The exact height of the osteotomy depended on the preoperative planning. The new leg axis was controlled with a long metal rod and an image intensifier. 

In Group 1, osteosynthesis was performed with the TomoFix™ implant (Synthes). In Group 2, osteosynthesis was performed with the LOQTEQ^®^ HTO plate (aap Implants, Berlin). The plates were temporarily fixed with K-wires approximately 1 cm distal to the proximal tibia joint line. Plate position was controlled with an imaging intensifier. Eight 5 mm locking screws were inserted for the TomoFix™ plate (4 screws in the proximal segment and 4 screws in the distal segment). Seven 4.5 mm locking screws were used for the LOQTEQ^®^ HTO plate (4 screws in the proximal segment and 3 screws in the distal segment). A torque screwdriver was used for both plates to finally tighten the locking screws. A compressive dressing was applied after skin closure.

### 2.3. Rehabilitation

All patients were mobilised with partial weight bearing with 10 kg for six weeks, and range of motion was not restricted. NSAIDs were not administered due to their inhibiting effect on bone healing. Stitches were removed postoperatively after 12 days. A low-molecular heparin was given for postoperative thrombosis prophylaxis.

### 2.4. Follow-Up Evaluation

All patients were examined by a fellowship-trained orthopaedic surgeon before surgery and postoperatively at 12 months and at 24 months using the Knee Osteoarthritis Outcome Score (KOOS) [17]. The primary outcome measure was the KOOS pain subscore. Secondary outcome criteria were other KOOS subscales symptoms, sports/recreational activities, activities of daily living, function [17], and activity evaluated with the Tegner scale. The KOOS was validated for the German language [18]. The KOOS subscales were assessed preoperatively (T0), 12 months postoperatively (T1), and 24 months postoperatively (T2). The Tegner scale was assessed at T0 and T2. Further secondary outcome criteria were degree of reduction, incision length (measured at the one-year follow up), radiological signs of implant failure on the 12-month radiographs, non-union rate, healing rate on the 12-month radiographs, and adverse effects.

Broken screws or plate were signs of implant failure. Non-union was defined as load-dependent pain at the osteotomy site in combination with insufficient bony healing in radiographs. Healing was assessed using the modified radiographic union score for tibial fractures (RUST) [19]. The original RUST assesses each cortex of a tibial fracture separately. Van Houten et al. [20] modified the RUST for the assessment of the healing of tibial osteotomies because the location of the fixation device makes scoring the anterior cortex on the lateral radiograph difficult for open-wedge HTOs. In the modified version, the lateral and medial cortex are scored on an AP radiograph, and only the posterior cortex is scored on the lateral radiograph. One point is given if a fracture line and no callus is visible; 2 points are given if a fracture line is visible, and callus is visible; and 3 points are given if a bridging callus and no evidence of a fracture line is visible. The scores for all cortices are summed for the total score. The minimum score of 3 indicates that the fracture is definitely not healed; the maximum score of 9 indicates that the fracture has healed [20].

Patients were asked at the 24-month follow up whether the implants had been removed and the reason for implant removal. The reports of implant removal procedures were screened for the time of surgery (incision to closure) and for complications such as jamming due to cold welding of the rectangular stable screws.

At the time of recruitment, the patients were told that they should contact the study office in case of an unexpected event. The charts were further reviewed for adverse effects such as delayed wound healing, haematoma, deep vein thrombosis, pulmonary embolism, and deep infection. The postoperative X-rays were screened for lateral hinge fracture and undisplaced lateral tibial plateau fracture. The adverse events were classified into three grades: Grade 1 (adverse events requiring no additional treatment), Grade 2 (adverse events requiring additional non-operative management), and Grade 3 (adverse events requiring additional or revision surgery).

### 2.5. Statistical Methods

Statistical analysis was performed by Dr Ulrike von Hehn at Medi Stat (Kiel, Germany). The Kolmogorov–Smirnov and Shapiro–Wilk tests were used to test the parameters for normal distribution. The Mann–Whitney U test was used for the results of the non-parametric parameters (KOOS and Tegner). A *t*-test was used for statistical analysis of the RUST. A chi-square test was used to examine the rate of implant removal and adverse effects. The significance level was set at *p* ≤ 0.05 for each test.

## 3. Results

### 3.1. Patients

Thirty-two patients were included in Group 1, and 35 patients were included in Group 2. One patient was lost for follow up in each group.

There was no significant difference in age and gender distribution between the two treatment groups (Table 2).

For radiographic evaluation, no significant differences were observed between the two cohorts for postoperative reduction. All findings are presented in Table 3.

Figure 2 illustrates the postoperative correction of the LOQTEQ^®^ HTO plate and TomoFix^TM^ plate.

### 3.2. Primary Outcome Measure

The KOOS pain subscale score increased significantly in both groups from T0 to T1. In Group 1 (TomoFix™), the KOOS pain increased from 44.6 (±23.9) at T0 to 73.1 (±25.4) at T1 and to 75.75 (±22.3) at T2. In Group 2 (LOQTEQ^®^ HTO plate), the KOOS pain increased from 49.2 (±22.2) to 72.3 (±23.0) at T1 and to 78.1 (±23.5) at T2. The difference between T0 and T1 and between T0 and T2 was statistically significant in both groups (Wilcoxon test for paired differences, *p* < 0.050). At no time was there a significant difference between Group 1 and Group 2 (U test, *p* ≥ 0.050).

### 3.3. Secondary Outcome Measures

KOOS subscores: function, symptoms, sports/recreational activities, and quality of life.

Figure 3 shows the results of the different KOOS subscores. Both groups showed a significant improvement over time between T0 and T1 and also between T0 and T2 for all KOOS subscores (for *p*-values, see figure legends). At no time, however, was there any evidence of a significant difference between the two groups (for *p*-values, see figure legends).

### 3.4. Tegner Scale

In Group 1 (TomoFix™), the activity measured with the Tegner scale (Figure 4) increased significantly from 2.7 (±1.6) at T0 to 3.4 (±1.4) at T2 (Wilcoxon test for paired differences, *p* < 0.05). In Group 2 (LOQTEQ^®^ HTO plate), the activity also increased significantly from 2.6 (±1.6) at T0 to 3.9 (±25.4) at T2 (Wilcoxon test for paired differences, *p* < 0.05). Although the increase in Tegner score in Group 2 was greater (1.2 in Group 2 vs. 0.7 in Group 1), this difference was not statistically significant (U test, *p* ≥ 0.05).

### 3.5. Radiological Signs of Implant Failure and Healing Rate on the 12-Month Radiographs

Radiological signs of implant failure were not observed in any case at the one-year X-ray, and no patient fulfilled the criteria for non-union. The RUST one year postoperatively was 8.5 (±0.7) in Group 1 and 8.6 (±0.7) in Group 2. This difference was not statistically significant (U test, *p* ≥ 0.05).

### 3.6. Rate, Cause, and Time of Implant Removal and Rate of Screws Jamming

Implant removal (Figure 5 and Figure 6) was performed at a mean of 16.8 months after surgery (minimum: 12 months, maximum: 20 months). In the TomoFix™ group, the plate was removed in 18 patients. In the LOQTEQ^®^ HTO group, the implant was removed in 21 patients. This difference was not statistically significant (chi-square test, *p* ≥ 0.05). Fifteen patients in the TomoFix™ group and 13 patients in the LOQTEQ^®^ HTO plate group reported discomfort as the cause of implant removal. This difference was also not statistically significant (chi-square test, *p* ≥ 0.05). The other patients reported no specific cause for the implant removal; they simply wanted it removed. In all patients, pain relief was reported. Screw jamming due to the cold welding of one or more rectangular screws was reported in eight surgical reports from the TomoFix™ group. No screw jamming was reported in the LOQTEQ^®^ HTO group. The surgical time for the implant removal procedure in the LOQTEQ^®^ HTO group was significantly shorter than in the TomoFix™ group (24.3 min ± 6.1 min vs. 36.8 min ± 14.9 min). This difference was statistically significant (U test, *p* ≥ 0.05).

### 3.7. Incision Length

The incisions were 6.5 cm (±1.2) in length in the LOQTEQ^®^ HTO group and 8.4 cm (±2.3) in the TomoFix™ group. This difference was statistically significant (U test, *p* ≥ 0.05).

### 3.8. Adverse Effects

Table 4 shows the absolute number of adverse effects. The most common adverse effect was a lateral hinge fracture followed by haematoma. Nearly all adverse effects were classified as Grade 1 (adverse events requiring no additional treatment, Grade 1). Only the deep venous thrombosis required additional prolonged, non-operative treatment with a low-molecular heparin (Grade 2). There was no significant difference in the frequency of adverse effects between the two treatment groups (chi-square test, *p* ≥ 0.05).

## 4. Discussion

The results of this study do not support our hypothesis. The KOOS pain scores after open-wedge HTO stabilised with the LOQTEQ^®^ HTO plate were not higher than the KOOS pain scores after HTO stabilised with the TomoFix™ (higher KOOS = lesser pain). It can be concluded that the smaller size of the plate does not contribute to less knee pain one or two years postoperatively. There was also no significant difference in the rate of implant removal or plate-associated discomfort, although there was a tendency for less discomfort in the LOQTEQ^®^ HTO group.

The other KOOS subscale scores, such as the scores for symptoms, function, sports/recreational activities, and quality of life, also increased over the course of the study, but there was no difference between the two plates tested. The KOOS is a patient-reported outcome measure, and this score was validated for the German language [18]. It has been shown that the KOOS is responsive for use in patients with knee osteoarthritis who are undergoing HTO [21]. The mean values of each of the KOOS domains in this study were above the threshold value for a large effect, suggesting clinically important changes [17]. This increase in the different KOOS domains after open-wedge HTO is in accordance with other clinical studies [1,4,22,23]. We believe that the large improvements in patient-reported outcomes observed during the course of this study highlight the potential benefit of surgically restoring neutral lower extremity alignment. Locking plates seem to be the ideal implants for medial open-wedge HTO in order to achieve good clinical outcomes [6,24].

Healing complications, such as non-union or implant failure, were not detected in either of the treatment groups. The TomoFix™ is known as an implant with a low non-union rate after open-wedge HTO [6,9,10,24,25]. In a retrospective study of 206 HTOs stabilised with the TomoFix™, the non-union rate was 4.9% [20]. Patients with known risk factors (smokers, obese patients, use of NSAIDs) [6] were included in this study. In the present study, these factors were exclusion criteria. These selection criteria might explain why no healing complications were observed in the present study. Higher rates of healing complications have been described for other implants. For example, Schröter et al. analysed outcomes after open-wedge HTO with the Position Plate^®^ (Aesqulap) in 35 patients [11]. A plate-related complication rate of 23% was found in that study [11]. Kyung et al. compared 25 open-wedge HTOs stabilised with the Position Plate^®^ and 25 cases where the TomoFix™ was used for osteosynthesis [10]. These authors found three cases of screw loosening and four cases of delayed union in the Position Plate^®^ group, while no plate-related complications were observed in the TomoFix™ group [10]. There were also six residual varus deformities in the Position Plate^®^ group and only one in the TomoFix™ group [10]. Shin et al. examined 50 patients after medial open-wedge HTO with the Decisive Wedge Locking Plate^®^ and 47 with the TomoFix™ [25]. In this study, the Decisive Wedge Locking Plate^®^ group had a higher rate of non-union (4%) compared to the TomoFix™ group (0%). The present study confirms the low rate of plate-related complications for the TomoFix™ implant and shows that the LOQTEQ^®^ HTO plate has a comparably low plate-associated complication rate.

There was also no difference in union, as evaluated with the modified RUST score, between the two treatment groups. The modified RUST score is a radiological instrument to evaluate the healing of osteotomies [20]. A minimum score of 3 indicates that the osteotomy is definitely not healed. A score of 7–9 indicates healing. Both implants tested in this study reached healing scores between 7 and 9. The good healing scores of both implants tested in the present study can be explained by their biomechanical properties [8]. Agneskirchner et al. [14] tested several implants for open-wedge high tibial osteotomy in a cadaveric model. The TomoFix^TM^ resisted the greatest amount of force in the single-load-to-failure tests in this study and more than twice the number of loading cycles in cyclic loading tests when compared with the short spacer plates (Puddu plate) [14]. A recently published biomechanical study showed that the fatigue strength of the LOQTEQ^®^ HTO plate was comparable to that of the TomoFix™ implant [15].

The seizure of locking screws is a commonly encountered clinical problem during implant removal of locking plates after completion of union [26]. This problem is said to occur in up to 20% of cases [27]. A previous biomechanical study suggested both over-tightening and cyclic loading as potential causes for screw seizure in locking plate implants [27]. This study found that both effects were less pronounced in locking screws with a conical head compared to a traditional screw head [27]. The results of the present study support these biomechanical findings. No screw jamming due to cold welding was observed in the LOQTEQ^®^ group, whereas cold welding was observed in eight cases at implant removal in the TomoFix™ group. The complication of screw jamming may have been a contributing factor to the longer surgical time for implant removal observed in the TomoFix™ group. Other factors were the longer incision and the number of screws (eight in the TomoFix™ group vs. seven in the LOQTEQ^®^ group).

Except for one case of thrombosis which required a higher dosage of low-molecular heparin, no other severe complications were found. No other complication required further treatment. These findings are in accordance with reports from the literature [28].

The most common complication that was not plate related in the present study was lateral hinge fracture, with six fractures in the LOQTEQ^®^ HTO plate group and seven fractures in the TomoFix™ group. The lateral hinge fractures did not change the postoperative treatment in the present study. The rate of lateral hinge fractures in a recent study was 22.6%, which is comparable to the rate found in the present study [29]. In this study, the TomoFix™ was used for osteosynthesis, and no radiologic or functional deterioration was observed in the group with lateral hinge fractures [29]. These authors concluded that a lateral hinge fracture does not affect outcomes after medial open-wedge high tibial osteotomy using a locked plate system [29]. The results of the present study confirm this.

The present study has some limitations. It is certainly a limitation that it was not a randomised study; however, randomised studies for surgical techniques are time consuming and also carry the risk of bias. Selection bias is a common limitation of randomised, controlled trials [30]. A post hoc power analysis revealed a power of 5.9%. The findings of these studies should, therefore, be generalised cautiously. No perioperative pain assessment was performed and nor was measurement of body mass index. In smaller patients, the plate size may impact highly. Further, the majority of patients were male. The homogeneity of the groups in the present study was ensured by applying strict selection criteria. This strict selection criteria, however, means that the rate of healing complications cannot be transferred to the normal population. The rate of thromboembolic events in both groups may be underestimated because systematic screening for thrombosis with Doppler ultrasound was not performed.

## 5. Conclusions

Comparison of two different plates (TomoFix™ vs. LOQTEQ^®^ HTO plate) following HTO and open-wedge osteotomy showed similar clinical outcomes and osseous healing, as well as complication rates. The disadvantages of the TomoFix™ plate are the longer incision length and the higher rate of screw jamming leading to complications at implant removal. It can, therefore, be concluded that the LOQTEQ^®^ HTO plate is a reliable and safe implant for the osteosynthesis of medial open-wedge osteotomies of the tibia. It could be used as an alternative for patients who care about the length of the incision. The smaller implant size could be advantageous for patients with a smaller body size.

## Figures and Tables

**Figure 1 jpm-13-00472-f001:**
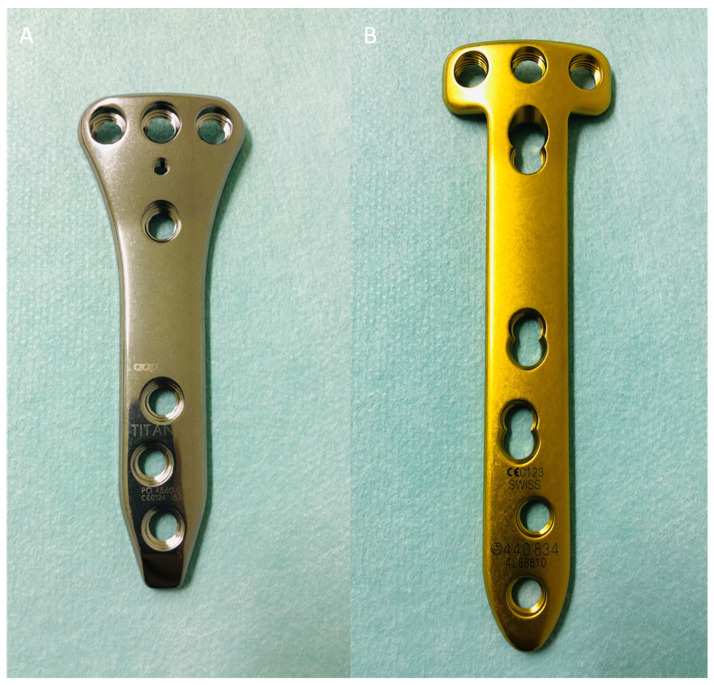
(**A**) LOQTEQ ^®^ HTO plate (aap Implants, Berlin). This osteosynthesis plate is an angular, stable internal plate fixator made of Ti_6_Al_4_V alloy. Absolute plate length is 90 mm. The plate has a trapezoid shape in its proximal third. Locking screw diameter is 4.5 mm. (**B**) TomoFix™ plate (Synthes, Switzerland). This osteosynthesis plate is an angular, stable internal plate fixator made of pure titanium. Absolute plate length is 120–150 mm. In its proximal third, the plate has a T shape. Locking screw diameter is 5.0 mm.

**Figure 2 jpm-13-00472-f002:**
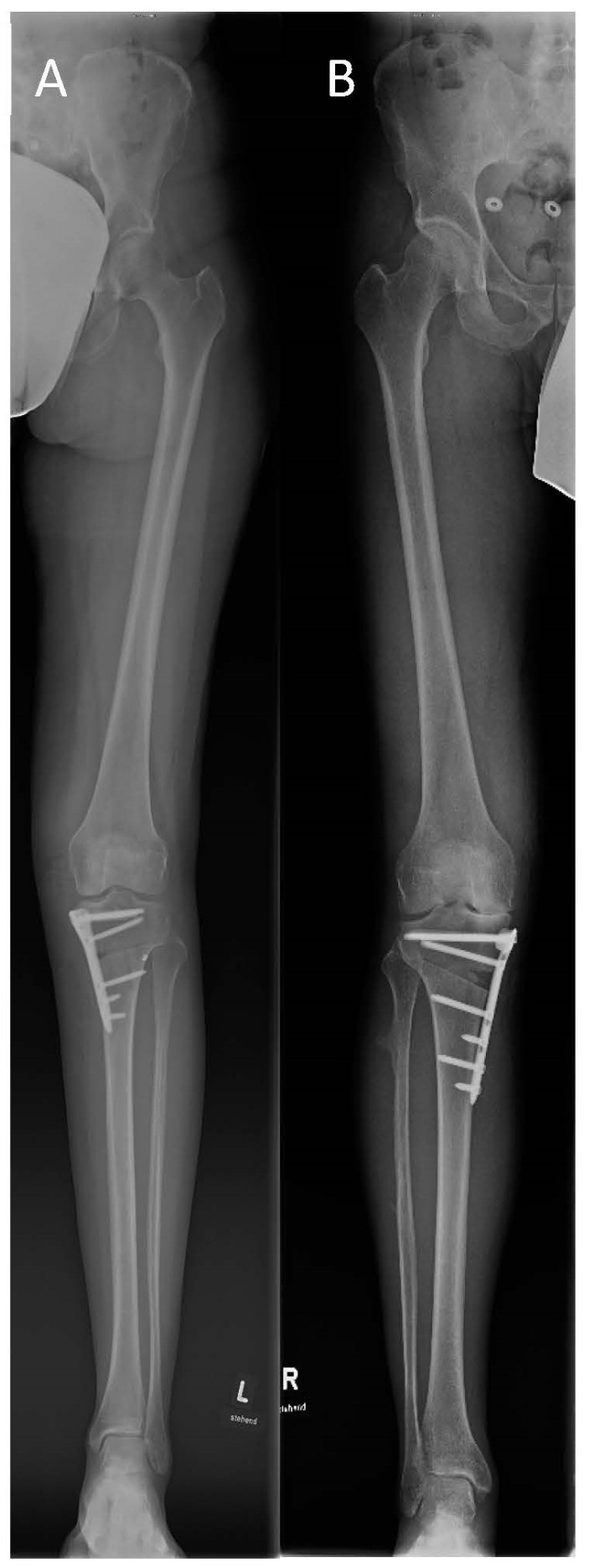
Full-length leg X-ray after HTO and open-wedge osteotomy using an LOQTEQ^®^ HTO plate (**A**) and a TomoFix^TM^ plate (**B**), respectively.

**Figure 3 jpm-13-00472-f003:**
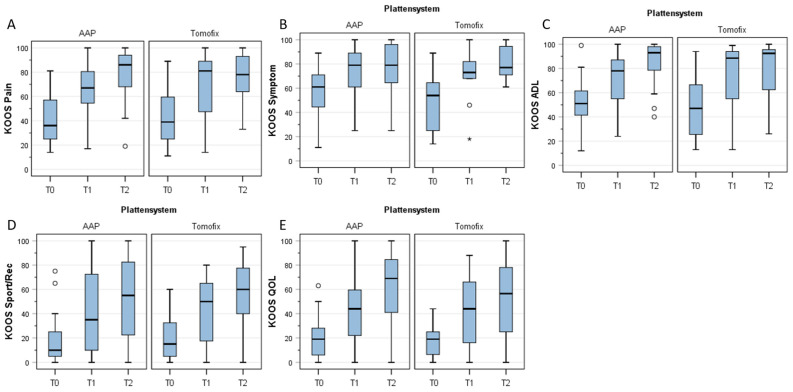
KOOS subscores, (**A**) pain, (**B**) symptoms, (**C**) function, (**D**) sports and recreational activities, and (**E**) quality of life. No differences were observed between the two cohorts.

**Figure 4 jpm-13-00472-f004:**
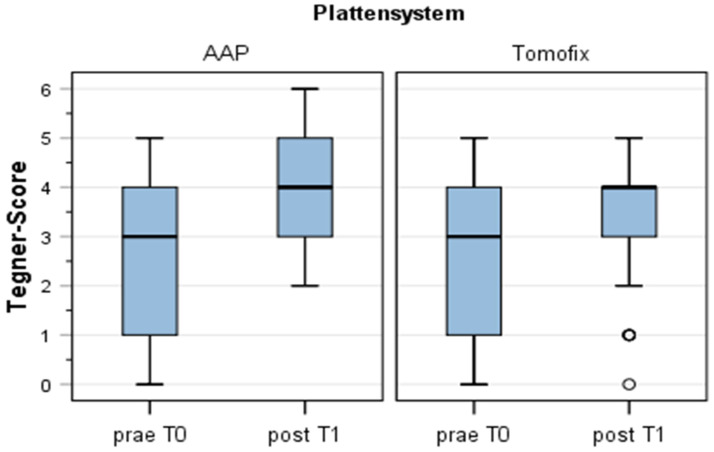
Tegner scale showed no significant difference between the different plates.

**Figure 5 jpm-13-00472-f005:**
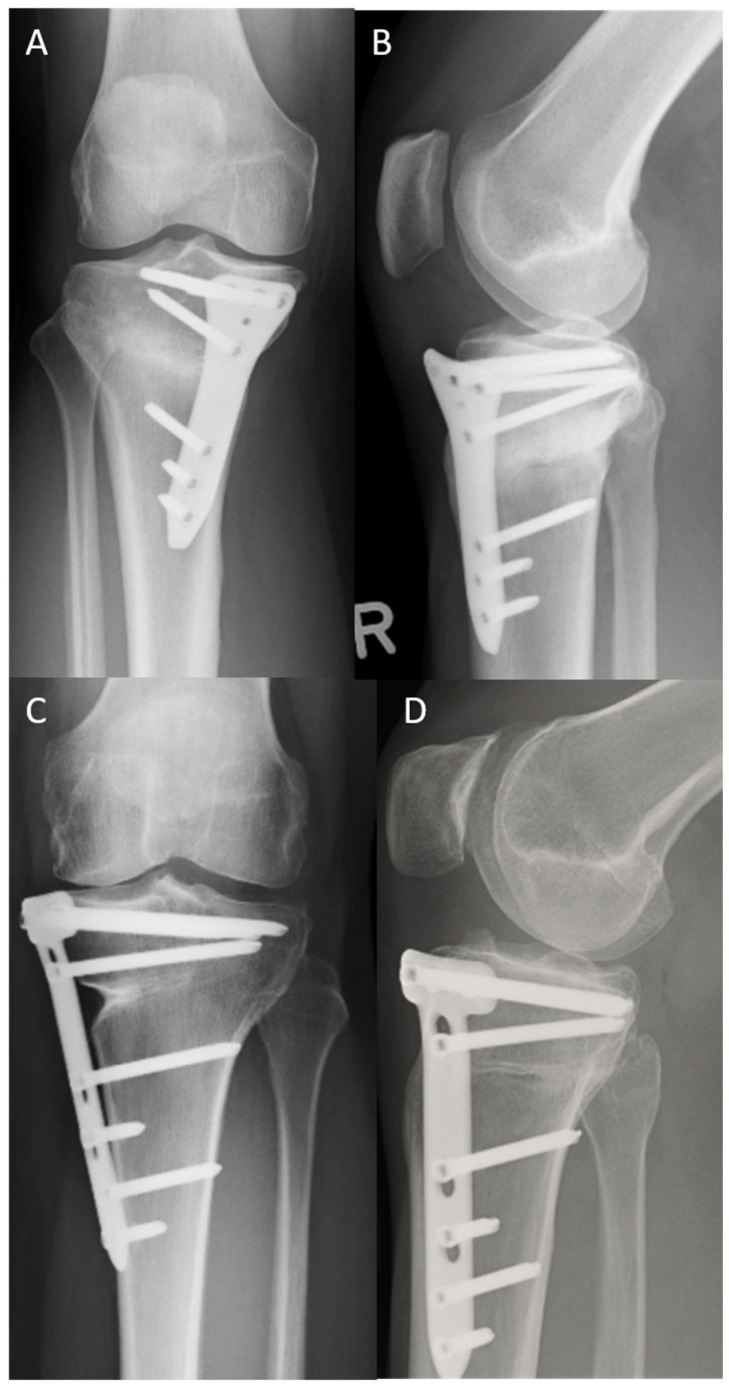
Post-operative X-rays before implant removal: (**A**) LOQTEQ^®^ HTO plate (aap Implants, Berlin), knee a.p.; (**B**) LOQTEQ^®^ HTO plate (aap Implants, Berlin), knee lateral; (**C**) TomoFix™ plate (Synthes, Switzerland) knee a.p.; (**D**) TomoFix™ plate (Synthes, Switzerland), knee lateral.

**Figure 6 jpm-13-00472-f006:**
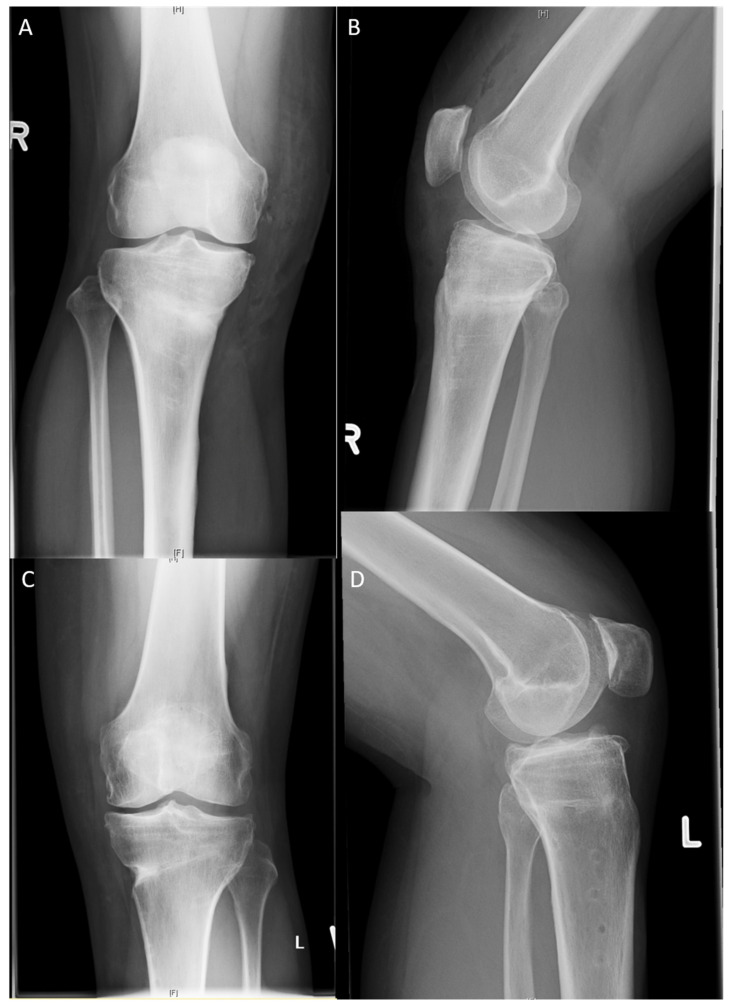
Post-operative X-rays after implant removal: (**A**) LOQTEQ^®^ HTO plate (aap Implants, Berlin), knee a.p.; (**B**) LOQTEQ^®^ HTO plate (aap Implants, Berlin), knee lateral; (**C**) TomoFix™ plate (Synthes, Switzerland), knee a.p.; (**D**) TomoFix™ plate (Synthes, Switzerland), knee lateral.

**Table 1 jpm-13-00472-t001:** Inclusion and exclusion criteria.

Inclusion Criteria	Exclusion Criteria
Medial osteoarthritis Grade 1–3 according to Kellgren and Lawrence; Varus deformity of between 5° and 10° (HKA); Willingness to undergo a high tibial osteotomy.	Smoker (more than one cigarette a day); BMI > 35; Medial meniscus root tear; Osteonecrosis of the medial femoral condyle; Additional anterior cruciate ligament plasty; Cartilage damage in the lateral compartment and/or in the patellofemoral joint of > than Grade 2 according to ICRS.

**Table 2 jpm-13-00472-t002:** Age and gender distribution.

	TomoFix™	LOQTEQ^®^ HTO Plate
Age	49.3 (±13.1)	50.2 (±10.4)
Gender	Female: 13 Male: 19	Female: 12 Male: 23

**Table 3 jpm-13-00472-t003:** Degree of reduction.

	LOQTEQ^®^ HTO Plate (aap Implants, Berlin)	TomoFix™ Plate (Synthes, Switzerland)	Overall	*p*-Value
Mechanical axis deviation in mm (preoperative)	18.5 ± 6.9	21.7 ± 10.0	20.1 ± 8.7	0.151
Mechanical axis deviation in mm (postoperative)	−8.0 ± 4.3	−7.7 ± 6.7	−7.8 ± 5.6	0.820
Hip–knee–ankle angle in degree (preoperative)	5.0 ± 1.7	6.1 ± 2.4	5.6 ± 2.2	0.057
Hip–knee–ankle angle in degree (postoperative)	−2.4 ± 1.2	−2.5 ± 1.7	−2.4 ± 1.4	0.765

**Table 4 jpm-13-00472-t004:** Adverse effects.

	TomoFix™ N: 32	LOQTEQ^®^ HTO Plate N: 35
Lateral hinge fractures	7	6
Undisplaced lateral tibial plateau fracture	0	0
Haematoma	1	2
Deep venous thrombosis	0	1
Non-union	0	0
Deep infection	0	0
Delayed wound healing	0	0

## Data Availability

All data are published in the manuscript.

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
