# Peer review of "Clinical Outcome after Medial Open-Wedge High Tibial Osteotomy: Comparison of Two Angular Stable Locking Plates—TomoFix™ versus LOQTEQ® HTO Plate"

_jpm, 2023, doi:10.3390/jpm13030472_

Round 1

Reviewer 1 Report

 Title: Clinical outcome after medial open wedge high tibial osteotomy: Comparison of two angular stable locking plates TomoFixTM versus LOQTEQ® HTO plate

Overall comments:

I would like to congratulate the authors for assessing a complex issue presented in clinic. It is a retrospective cohort study, with data collection completed prospectively. These two implants have been validated either clinically or biomechanically with good results. Based on the results of the study, both implants can be used clinically. However, there are several factors that the authors failed to address in the study. Most notably, there were no preoperative or postoperative deformity measurements presented. I believe this is the most important concept behind HTO in which deformity correction leads to improve clinical symptoms. Union rate should not be the primary outcome. It should be angle correction comparison as well as clinical scores. In addition, there are several comments and changes that I recommend for the authors. 

Decision:

Reject due to insufficient information provided 

Abstract:

Information provided is appropriate and in line with the study.

Introduction:

Line 75-76 I think hypothesis of better pain relief due to smaller implant size is a correct hypothesis. I believe during early postoperative period the pain is mostly due to the osteotomy and soft tissue dissection. It may decrease long-term plate related issues. This statement is somewhat misleading.

Methods:

Line 83-90 and table 2. I would like to know more about the patient baseline characteristics. It seems like you only included age and gender. I would also like to know the OA grading, deformity angle such as medial proximal tibia angle, if leg length discrepancy was apparent, if there were any concomitant deformities. As of now, it is insufficient information to determine if these patient population is the same.

Line 98-102 was a bone graft used to fill the defect? If so, what type? Also, was a PSI guide used after preoperative planning? 

Line 98-99 One difference I think is incision length. So I would also mention this in my differences between the two surgeries. Also, was a tourniquet applied during the surgery?

Line 98 who performed the surgeries? Was it one or multiple surgeons?

Line 100-101 I am unfamiliar with this software. Does it only create an osteotomy planning or it can also show the plate with the osteotomy? 

Line 111 torn meniscus was removed. I would assume this excludes MM root tear?

Line 148 Was there any pain scale assessed in the perioperative period? For instance pain at immediate postop and early postop with VAS scale etc.

Line 148 who performed the assessments? The surgeon that performed the surgery? 

Line 179-184

Was a power analysis completed? To me the first thing I consider when a study does not have significance is whether it was underpowered?

Line 143-177 With no preoperative angles information, you will not be able to assess for postoperative angle correction. The primary goal of high tibial osteotomy is to allow for correction of varus angle deformities which can change the biomechanical stress from this. In addition, there was no measurement of intraoperative incision length. When was union achieved in both groups? All parameters assessed were focused on union and follow-up at 12 and 24 months. I think there are significant information that we are missing in preoperative evaluation and postoperative assessment. Also was tibia slope corrected during the procedure?

Results:

Line 187-190 and table 2. This is inadequate information regarding patient baseline characteristics. 

Table 2 Also, why was there more male patients in the LOQTEQ group? In my experience, the patients that had most implant related issues were female patients due to their body habitus and lack of muscle coverage. 

Line 192-198 p value presentation. I think p-value should be given as an exact value to the tenth or hundredth decimal value.  I also think all of the values for pain and differences should be presented in a table with figures and words supplementing for clariy purposes.

Line 228-240: Implant removal was indicated for patients that had pain and discomfort. Was the pain resolved after pain removal? What was the percentage of people did reported improvement after removal?

Line 238-239 how was surgical time assessed? One screw difference should not be 12 minutes difference

255-257 how was incision length assessed? This was not mentioned in methods. Was this before or after removal of implants?

Discussion:

Line 267-270 with no power determined I think any conclusions or significance is hard to be drawn from the study

Line 283-284 Since there was no data presented for deformity correction, this study cannot justify this point

Line 336-341 can we classify and elaborate more on the type of hinge fractures in this study? How was it managed? What was the cause and was it noted intraoperatively?

Line 307-308 I would not say bone healing, RUST only can tell us union at what time point. There were no serial CT scans or different time point x-rays to confirm healing is the same in these two groups 

Line 344: the study only looked at union of the osteotomy, which I believe is important but not the most important factor. No assessment of correction of deformity angle was mentioned.

Conclusions:

Line 353-362 Conclusion of the study is inappropriate. The study design was used to compare two plates. It doesn’t justify HTO or open wedge technique is a safe and efficient procedure. Moreover, the authors failed to conclude any deformity angles in this study hence it is does just justify it as a procedure for varus deformity.

Moreover, incision length was not a primary outcome and is not clearly described on how it was measured and assessed. 

Lastly, we cannot infer from study results that smaller implant size could be advantageous for patients in smaller body size. There was never a mention of patient body habitus in this study or BMI or calf size etc. 

Author Response

Dear Reviewer,

thank you so much. We made all effort to comply with your comments.

Many thanks.

Reviewer 2 Report

The paper is well-written, it provides good introduction, conducted methods are described in detail and results are clearly presented. However, I would suggest to improve the quality and structure of figures to increase the quality of the paper:

- please reorganize Figures 4 and 5 to make them compact and make sure that the gaps between different subfigures in this Figures are the same

- please add appropriate letters (A-E) on each subfigure in Figures 1, 2, 3, 4 and 5 

- please extend the description of Figures 2 and 3

I believe that the present paper will be of interest to the broad audience of Journal of Personalized Medicine.

Author Response

(The authors gave the same response as above.)

Round 2

Reviewer 1 Report

I thank the authors for completing a wonderful response and to me, the manuscript is thorough and complete. I would accept the manuscript with minor editing in English.